

# An earliest Triassic age for *Tasmaniolimulus* and comments on synchrotron tomography of Gondwanan horseshoe crabs

Russell D.C. Bicknell[1], Patrick M. Smith[2,3], Tom Brougham[1] and Joseph J. Bevitt[4]

[1] University of New England, Armidale, Australia
[2] Australian Museum Research Institute, Sydney, Australia
[3] Macquarie University, Sydney, Australia
[4] Australian Nuclear Science and Technology Organisation, Sydney, Australia

## ABSTRACT

Constraining the timing of morphological innovations within xiphosurid evolution is central for understanding when and how such a long-lived group exploited vacant ecological niches over the majority of the Phanerozoic. To expand the knowledge on the evolution of select xiphosurid forms, we reconsider the four Australian taxa: *Austrolimulus fletcheri*, *Dubbolimulus peetae*, *Tasmaniolimulus patersoni*, and *Victalimulus mcqueeni*. In revisiting these taxa, we determine that, contrary to previous suggestion, *T. patersoni* arose after the Permian and the origin of over-developed genal spine structures within Austrolimulidae is exclusive to the Triassic. To increase the availability of morphological data pertaining to these unique forms, we also examined the holotypes of the four xiphosurids using synchrotron radiation X-ray tomography (SRXT). Such non-destructive, *in situ* imaging of palaeontological specimens can aid in the identification of novel morphological data by obviating the need for potentially extensive preparation of fossils from the surrounding rock matrix. This is particularly important for rare and/or delicate holotypes. Here, SRXT was used to emphasize *A. fletcheri* and *T. patersoni* cardiac lobe morphologies and illustrate aspects of the *V. mcqueeni* thoracetronic doublure, appendage impressions, and moveable spine notches. Unfortunately, the strongly compacted *D. peetae* precluded the identification of any internal structures, but appendage impressions were observed. The application of computational fluid dynamics to high-resolution 3D reconstructions are proposed to understand the hydrodynamic properties of divergent genal spine morphologies of austrolimulid xiphosurids.

Corresponding author
Russell D.C. Bicknell,
rdcbicknell@gmail.com

## INTRODUCTION

The increasing availability of three-dimensional (3D) imaging techniques in the preceding two decades has revolutionised the acquisition of morphological data from both biological (*Hita Garcia et al., 2017*; *Parapar et al., 2017*; *Landschoff et al., 2018*; *Marcondes Machado,*

*Passos & Giribet, 2019*; *Raymond et al., 2019*) and palaeontological specimens (*Sutton, 2008*; *Pardo & Anderson, 2016*; *Liu, Rühr & Wesener, 2017*; *Liu et al., 2019*; *Forel, Poulet-Crovisier & Korat, 2021*). Traditional lab-based micro-computed tomography (CT), along with more sophisticated synchrotron radiation X-ray tomography (SRXT) and neutron micro-tomography (NCT) have permitted non-destructive visualisation of previously unknown and inaccessible morphological features for taxa across all of Metazoa (*Donoghue et al., 2006*; *Tafforeau et al., 2006*; *Sutton, 2008*; *Metscher, 2009*; *Motchurova-Dekova & Harper, 2010*; *Faulwetter et al., 2013*; *Faulwetter et al., 2014*; *Herrera et al., 2020*; *Snyder et al., 2020*). This precludes the need for physical dissection and/or preparation of specimens, which is relevant when describing structures from rare or fragile material (*e.g.*, *Metscher, 2009*; *Haszprunar et al., 2011*; *Deans et al., 2012*; *Beutel et al., 2019*; *Willsch et al., 2020*; *MacDougall et al., 2021*; *Stillwell et al., 2020*). In palaeontology, 3D data has been used widely in the visualisation of fossils preserved in amber (*Lak et al., 2008*; *Perrichot et al., 2008*; *Riedel et al., 2012*; *Xing et al., 2016a*; *Xing et al., 2016b*; *Xing et al., 2018*; *Daza et al., 2020*; *Bolet et al., 2021*) and also in the examination of fossils that are still surrounded in their original rock matrix (*Moreau et al., 2014*; *Schwarzhans et al., 2018*; *Reid et al., 2019*; *Mayr et al., 2020*).

Research into fossil arthropods has benefitted greatly from the availability of non-destructive 3D imaging techniques (*Deans et al., 2012*; *Liu et al., 2016*; *Liu et al., 2020*; *Hegna, Martin & Darroch, 2017*; *Wesener, 2019*; *Zhai et al., 2019a*; *Zhai et al., 2019b*; *Liu et al., 2020*), particularly the diverse array of insects preserved within resins (*Tafforeau et al., 2006*; *Lak et al., 2008*; *Pohl et al., 2010*; *Henderickx, Tafforeau & Soriano, 2012*; *Riedel et al., 2012*). In stark contrast, extinct members of Xiphosurida (*i.e.,* horseshoe crabs) have received comparatively limited 3D examination. The anatomy of two extant xiphosurids, the American horseshoe crab—*Limulus polyphemus* (*Linnaeus, 1758*)—and the mangrove horseshoe crab—*Carcinoscorpius rotundicauda* (*Latreille, 1802*)—has been documented using micro-CT (*Göpel & Wirkner, 2015*; *Bicknell et al., 2018a*; *Bicknell et al., 2018b*; *Bicknell et al., 2021b*; *Bicknell, Melzer & Schmidt, 2021*). Magnetic resonance imaging has also been used in studies of the Japanese horseshoe crab—*Tachypleus tridentatus* (*Leach, 1819*) (*Kutara, Une & Fujita, 2019*; *Yuen, Kwok & Kim, 2019*). However, as *Bicknell & Pates (2020)* highlighted, there are over 80 extinct xiphosurids that have not been documented or rendered in 3D and most 3D data collected from fossil xiphosurids have been surface scans (*Schimpf et al., 2017*), with other applications including stereo imaging (*Haug et al., 2012*; *Haug & Rötzer, 2018*; *Haug & Haug, 2020*). A recent study combined CT and computed laminography (*Zuber et al., 2017*) to image *Limulitella Størmer, 1952* from the Winterswijk quarry complex, Middle Triassic (Anisian) Vossenfeld Formation, Netherlands (*Klompmaker & Fraaije, 2011*; *Klein, 2012*; *Sander et al., 2016*; *Zuber et al., 2017*). These techniques revealed morphological information that was not visible due to the compression and preservation of the specimen. However, no other fossil xiphosurids have been examined using comparable methods. Here we address this lack of data by presenting the first application of SRXT to holotypes of four Australian xiphosurids. In doing so, we also reconsider the temporal range of these four taxa. This revision uncovers

a younger age for one genus, pushing the record of Austrolimulidae in Australia to the Triassic.

## METHODS

We examined the four species of Xiphosurida known from Australia using SRXT: *Austrolimulus fletcheri Riek, 1955* from the Hawkesbury Sandstone (Middle Triassic, Anisian), New South Wales (NSW); *Dubbolimulus peetae Pickett, 1984* from the Napperby Formation (Middle Triassic, Anisian), NSW; *Tasmaniolimulus patersoni Bicknell, 2019* from the Jackey Shale (Early Triassic, Induan), Tasmania; and *Victalimulus mcqueeni Riek & Gill, 1971* from Koonwarra Fossil Bed (Early Cretaceous, Aptian), Victoria. All four species fall within the xiphosurid groups Limulidae and Austrolimulidae (*Bicknell, 2019*; *Bicknell et al., 2021a*; *Lamsdell, 2021*). Given advances in the stratigraphic literature since the initial descriptions of these four forms, we conducted a literature review and present a thorough geological contextualisation for each taxon.

Non-destructive X-ray microtomographic measurements were conducted using the Imaging and Medical Beamline at the Australian Nuclear Science and Technology Organisation's (ANSTO) Australian Synchrotron, Clayton, Victoria, Australia.

A monochromatic beam energy of 70 keV was used for *Dubbolimulus peetae* and *Victalimulus mcqueeni*, with a sample-to-detector distance of 500 mm. X-rays were converted to visible photons and detected using the "Ruby detector", a 20 $\mu$m thick Gadox/CsI(Tl)/CdWO$_4$ scintillator screen coupled with a PCO.edge sCMOS camera (16-bit, 2,560 $\times$ 2,160 pixels) and a Nikon Makro Planar 50 mm lens to achieve a pixel size of 24.8 $\times$ 24.8 $\mu$m. A total of 1800 equal angle shadow-radiographs were obtained (*i.e.*, one radiograph every 0.10°) with an exposure length of 0.070 s each as the samples were continuously rotated 180° about their vertical axes. Due to the restricted beam height and field-of-view, this radiograph capture procedure was repeated after lowering the specimen with respect to the beam after a full rotation. This produced a series of overlapping vertical radiographs capturing the full height of each specimen. These were then stitched together into a single set of radiographs prior to reconstruction into 3D volumes. For *V. mcqueeni* the reconstructed data was binned to voxels of 49.6 $\mu$m for visualisation. *Tasmaniolimulus patersoni* and *Austrolimulus fletcheri* were similarly scanned with a pixel size of 40.29 $\times$ 40.29 $\mu$m. An incident monochromatic beam energy of 80 keV was used for *T. patersoni* and a broad range of higher energy X-rays (pink beam, peak energy of 220 keV) was used for *A. fletcheri* due to the high attenuation of available monochromatic X-rays.

The raw 16-bit radiographs were normalised relative to the beam calibration files, stitched using the in-house software IMBL Stitch, and reconstructed with CSIRO's X-TRACT (*Gureyev et al., 2011*) software available on Australian Synchrotron Computing Infrastructure (ASCI). The filtered-back projection reconstruction method was used to form a 16-bit, 3D volume image of the sample.

The reconstructed slices for each fossil were imported into Mimics version 23.0 (Materialise, Leuven, Belgium) and digitally prepared. Any artefacts in the tomographic slices were removed using the 'Segmenting' tool and the remaining components (fossil

and matrix) were segmented out and converted to .STL files in Mimics, and imported into Geomagic Studio (3D Systems, North Carolina, USA) to be smoothed. The smoothed .STL files were used to generate 3D PDFs using Terta4D (Adobe Systems; see Figs. S1– S4 found at 10.17605/OSF.IO/AT528). Lighting used in the 3D PDFs was Computer-Aided Design optimised to showcase features prominently and without shadowing. Raw radiograph data associated with this research has been uploaded to MorphoSource (https://www.morphosource.org/projects/000380648). Photographs of each specimen were taken under LED lighting either by the authors or by collection managers for overall comparison to the 3D reconstructions. A note here must be made to the use of stereo-photographs. This imaging technique has effectively been used to illustrate fossil arthropods (*Haug et al., 2009*; *Haug, Martin & Haug, 2015*; *Haug, Müller & Haug, 2019*; *Haug, 2020*) and particularly fossil xiphosurids (*Haug et al., 2012*; *Haug & Rötzer, 2018*; *Haug & Haug, 2020*). This has been especially informative when specimens are dorsoventrally compressed and may have revealed more structures than the LED lighting photography conducted here. However, as the focus of this research was on the synchrotron scanning and digitisation of the holotypes, we did not apply this method here. Nonetheless, future work on fossil xiphosurid anatomy should consider gathering stereo images for comparative purposes.

Three-dimensional models can also be produced using photogrammetry. This method is particularly useful for illustrating overall specimen morphology and models are cost-effective to produce (*Falkingham, 2012*; *Cunningham, 2021*). However, photogrammetry cannot be used to gather data on internal structures—one of the main focuses here. As such, we did not explore the application here. Regardless, photogrammetry should be considered for future research interested in overall 3D morphology of horseshoe crabs.

## Geological context

The oldest Australian xiphosurid, *Tasmaniolimulus patersoni*, was found in the Jackey Shale of the Upper Parmeener Supergroup, Tasmania (*Bicknell, 2019*). This formation is largely composed of cross-bedded quartz and feldspathic sandstones, laminated dark grey shales and thin coal lenses (*Pike, 1973*). Stratigraphically, the fossil was located near the very top of the formation, ~3 m below the base of the overlying Ross Formation, exposed alongside a cliff on the Poatina Highway (41°48′05″S, 146°53′06″E; *Ewington, Clarke & Banks, 1989*; *Bicknell, 2019*). Based on the lithology, the unit likely represents deposition of lake and river sediments in a non-marine swamp with limited coastal influence (*Banks, 1973*; *Ewington, Clarke & Banks, 1989*). While the Jackey Shale at the stratigraphic level of the collection locality lacks age-diagnostic fossils, palynomorphs from other, temporally contiguous sites can be assigned to the *Protohaploxypinus microcorpus* Zone, equivalent to upper APP6 (see *Price, 1997*) and restricted to the Griesbachian substage, early Induan (Early Triassic) based on previous studies in the Sydney Basin (*Laurie et al., 2016*; *Mays et al., 2020*). This contradicts previous interpretations of latest Permian that used now outdated chronostratigraphic ages for this palynomorph zone. An Early Triassic age is further supported by the vertebrate fauna and macro- and microflora of the *Protohaploxypinus samoilovichii* Zone from the overlying Ross Formation which pertains to the younger Smithian substage of the Olenekian (Early Triassic; *Forsyth, 1984*). The presence of

abundant latest Permian macroflora at stratigraphic levels below the level of *T. patersoni* in the Jackey Shale does suggest that, at least at some locations, the formation does extend into the latest Permian (*Ewington, Clarke & Banks, 1989*). Nonetheless, given the high stratigraphic position of *T. patersoni*, it appears more likely that this specimen is of Early Triassic age.

Slightly younger is *Dubbolimulus peetae*, which was collected from the Napperby Formation (previously the "Ballimore Formation") of the Gunnedah Basin in central New South Wales (*Pickett, 1984*). The only known specimen, with an associated counterpart, was found just south of Western Plains Zoo, Dubbo (at approximately 32°17′30.8″S 148°34′35.8″E). The Napperby Formation consists of white, fine–medium grain, quartz-rich, ferruginous sandstone with occasional cross bedding. Thin horizons of grey to red-brown shale and minor conglomerate lenses are interbedded with this sandstone. The stratigraphic horizon within which the specimen was found is a red-brown, slightly micaceous shale. This lithology indicates a high-energy braided river system or lacustrine deposit (*Tadros, 1993*), possibly part of the same Triassic delta system that continues into the Sydney Basin to the east. The finer grained shale horizons likely represent lower-energy conditions which presumably occurred in quiet, cut-off river channels or small ponds. The possible presence of acritarchs (McMinn, unpublished data, 1982; Early Permian-Early Jurassic palynology of DM Mirrie DDH 1, northwest of Dunedoo. Geological Survey of New South Wales, Report GS1982/289) suggest the unit may have experienced a slight coastal influence occasionally. A diverse macroflora assemblage has been described from both the fossil site itself (*Pickett, 1984*) and a nearby locality (*Holmes, 1982*) which broadly correlate to the *Dicroidium zuberi* Zone (*Helby, 1973*; *Helby, Morgan & Partridge, 1987*; *Retallack, 1977*; *Retallack, 1980*; *Helby, Morgan & Partridge, 1987*) of the Anisian (earliest Middle Triassic) in the Sydney Basin. Palynomorphs from core within the Dubbo area, at Mirrie DOH I (McMinn, unpublished data, 1982; Early Permian-Early Jurassic palynology of DM Mirrie DDH 1, northwest of Dunedoo. Geological Survey of New South Wales, Report GS1982/289) and Pibbon DOH 1 (McMinn, unpublished data, 1984; Palynology of DM Pibbon DDH 1, Goulburn River-Binnaway area. Geological Survey of New South Wales, Report 84/4, GS1984/052), support this age interpretation with placement in the *Aratrisporites parvispinosus* Zone which correlates to the middle to upper *Dicroidium zuberi* Zone (*Young & Laurie, 1966*). A middle *D. zuberi* Zone stratigraphic position, which indicates an earliest Anisian age, is most likely given palynomorphs from other locations in the Gunnedah Basin, which suggest an age range between the upper *Aratrisporites tenuispinosus* Zone and lower *Aratrisporites parvispinosus* Zone.

Of a similar age is *Austrolimulus fletcheri*, from Beacon Hill Quarry, near the suburb of Brookvale, Sydney, New South Wales (*Riek, 1955*). The exact co-ordinates of the original collection site are unknown, but are considered to be 33°45′11.2″S, 151°15′55.5″E; the location of the original quarry. The specimen originates from a 8 m thick shale lens in the Hawkesbury Sandstone. This lens mostly consists of numerous thin, recessive, grey-red mudrock laminations with little bioturbation (*Webby, 1970*) and small amounts of rippling (*Herbert, 1983*). Overall, the Hawkesbury Sandstone was likely formed in a vast coastal floodplain made up of high energy braided rivers, scour channels, lakes, and sand dunes

(*Conaghan, 1980* and references therein). Shale lenses, like those at the *A. fletcheri* site, likely represent lower-energy regimes consisting of shallow water bodies disconnected from a main river channel as isolated shallow pools of water (*Herbert, 1980*; *Herbert, 1997*; *Rust & Jones, 1987*). None of the diverse fossil fauna and flora found at Brookvale (see *Bicknell & Smith, 2021* for a recent overview) are diagnostic for relative age estimation. However, the Hawkesbury Sandstone is well constrained within the *Aratrisporites parvispinosus* Zone and upper *Dicroidium zuberi* Zone based on palynomorphs and macroflora (*Helby, 1973*; *Retallack, 1977*; *Retallack, 1980*; *Helby, Morgan & Partridge, 1987*). Similar to the Napperby Formation, this places it within the Anisian (earliest Middle Triassic), likely the earliest Anisian. Recent high-precision U-Pb CA-TIMS obtained from the Garie Formation, which underlies the Newport Formation and succeeding Hawkesbury Sandstone, is dated to the latest Olenekian (248.23 ± 0.13 Ma and 247.87 ± 0.11 Ma; *Metcalfe et al., 2015*). This further supports an Anisian age for the Hawkesbury Sandstone as there is an unconformity in the Sydney Basin between Newport Formation and Hawkesbury Sandstone (*Helby, 1973*; *Herbert, 1980*).

*Victalimulus mcqueeni* from Koonwarra Fossil Bed of the Strzelecki Group (*Riek & Gill, 1971*), is the youngest xiphosurid known from Australia. A single partial specimen was found at a road cutting along the South Gippsland Highway, approximately 2.4 km east of Koonwarra (38°33′48.9″S 145°57′33.9″E). The unit at this location consists of a thick (∼7–8 m) lower and upper feldspathic sandstone bracketing a grey-green, fossiliferous mudstone (*Waldman, 1971*; *Jell & Roberts, 1986*). The mudstone is made up of extremely fine alternating layers of a clay- and silt-dominated matrix. A freshwater lacustrine environment was originally suggested for the Koonwarra Fossil Bed, with the finely laminated mudstones representing a rhythmic varve formed under freezing conditions (*Waldman, 1971*; *Waldman, 1973*; *Waldman, 1984*). However, the highly diverse fossil fauna and flora (see overview in *Poropat et al., 2018*), instead suggests a cold, but not freezing, swamp or a lacustrine environment with seasonal flooding causing overbank-type deposits (*Douglas & Williams, 1982*; *Jell & Roberts, 1986*). Presence of the palynomorphs *Clavatipollenite hughesii Couper, 1957* and *Foraminisporis asymmetricus Dettmann, 1963* from the Koonwarra Fossil Bed, and absence of other palynomorphs from younger zones, indicate an age within the Upper *Cyclosporites hughesii* subzone (*Jell & Roberts, 1986*; *Seegets-Villiers & Wagstaff, 2016*; *Korasidis & Wagstaff, 2020*; *Wagstaff et al., 2020*). This places the unit entirely within the Aptian Stage (Early Cretaceous). Fission track dating of volcanoclastic sediments in the Koonwarra Fossil Beds suggests an age of 118 ± 5–115 ± 6 Ma, which correlates to the mid-Aptian (*Gleadow & Duddy, 1980*; *Lindsay, 1982*).

## RESULTS

The reconstructed tomographic volumes emphasized morphological information that was less evident under visible wavelengths. The density of the matrix surrounding *Austrolimulus fletcheri* precluded the unambiguous identification of many internal structures (Fig. 1). However, the cardiac lobe can be more readily distinguished in the reconstructed volume

and more depth is observed than exposed on the dorsal surface of the fossil (Fig. 1C). Furthermore, the composition of the genal spines is less dense than the prosoma, suggesting a limited portion of the spine was less sclerotised (Fig. 1D). *Dubbolimulus peetae* shows no evidence of preserved internal structures. The limited record of anatomical features reflects the strong dorsoventral compression of the specimen (Fig. 2). However, examination of the surface reconstruction reveals impression of the walking legs. These structures are also observed under LED light (Fig. 2A). The cardiac lobe of *Tasmaniolimulus patersoni* is the most prominent feature visible in the reconstruction (Fig. 3). This structure is observed at different slices in the reconstruction, illustrating the pronounced nature of the cardiac lobe. Finally, the reconstruction of *Victalimulus mcqueeni* reveals the most anatomical data of the four specimens. There is clear evidence for the thoracetronic doublure, fixed spines, moveable spine notches, and appendage impressions, as noted by *Riek & Gill (1971)* (Fig. 4). The cardiac lobe is not as pronounced as in *A. fletcheri* and *T. patersoni*, reflecting the more compressed nature of *V. mcqueeni*.

## DISCUSSION

### Age of *Tasmaniolimulus patersoni*

The revised earliest Triassic age of *Tasmaniolimulus patersoni* has important implications for the timing of morphological innovation within Austrolimulidae. *Tasmaniolimulus patersoni* was originally considered to be of latest Permian age (*Ewington, Clarke & Banks, 1989*; *Lerner, Lucas & Lockley, 2017*; *Bicknell, 2019*; *Lamsdell, 2020*). This age indicated that the first appearance of hypertrophied genal spines in Austrolimulidae was before the end-Permian extinction (*Bicknell, Naugolnykh & Brougham, 2020*). However, the revised date shifts the first appearance of this trait to the earliest Triassic. Furthermore, *T. patersoni* is now either the oldest Triassic austrolimulid, or contemporaneous with *Vaderlimulus tricki Lerner, Lucas & Lockley, 2017* and *Psammolimulus gottingensis Lange, 1923*—taxa that all have overdeveloped genal spine morphologies (*Meischner, 1962*; *Lerner, Lucas & Lockley, 2017*; *Bicknell, Hecker & Heyng, 2021*).

### Comments on application of synchrotron tomography to the study of fossil xiphosurids

The SRXT examination of the Australian xiphosurid fossils did not reveal extensive novel anatomy, nor traces of soft tissues. The aforementioned specimens were preserved primarily in sand- and siltstones which limits the preservation potential of fine, delicate structures. This is in contrast to the tomographic and laminographic reconstructions of the xiphosurid described by *Zuber et al. (2017)* and which was preserved in fine grained, Muschelkalk-type limestones. These sediments tend to preserve soft-bodied anatomical details in exceptional detail (*Vía, De Villata & Esteban Cerdá, 1977*; *Briggs & Gall, 1990*; *Cartañài Martí, 1994*; *Klug, Hagdorn & Montenari, 2005*). Nonetheless, non-destructive three-dimensional imaging using SRXT will likely continue to play a role in anatomical studies of fossil xiphosurids, following the rapid adoption of this imaging modality across palaeontology. Furthermore, NCT is being used more commonly in palaeontology, owing to the ability of neutrons to penetrate through typically radiopaque minerals

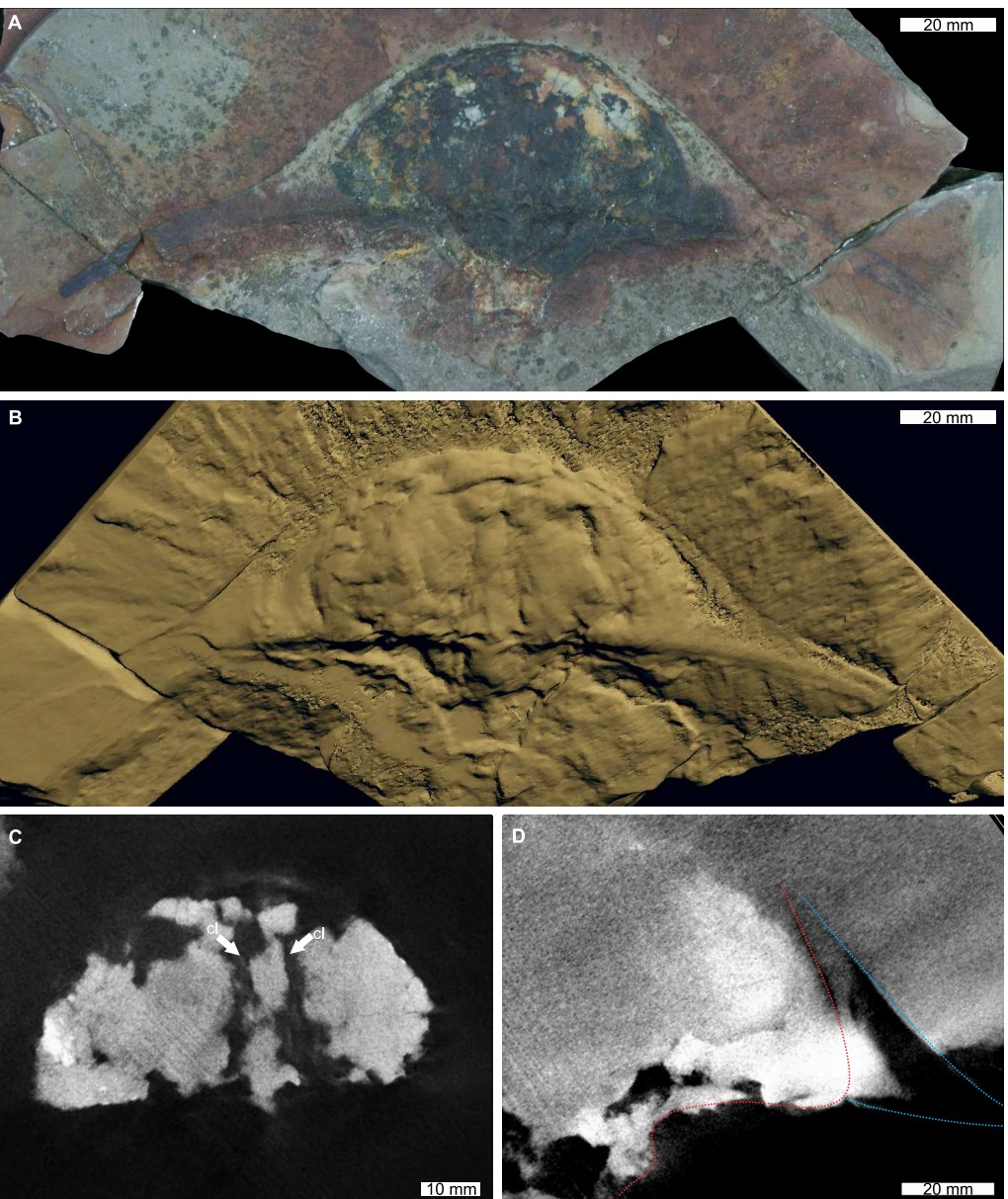

**Figure 1** *Austrolimulus fletcheri* **from the Hawkesbury Sandstone (Middle Triassic, Anisian). AM F38275 counterpart of holotype.** (A) Specimen under LED light. (B) 3D reconstruction of specimen, see Fig. S1. (C) X-ray tomographic slice showing pronounced cardiac lobe (white arrows). (D) X-ray tomographic slice showing difference in density between prosoma (red dotted line) and hypertrophied genal spine (blue lines). Abbreviation: cl, cardiac lobe. Image credit: (A) Joshua White. 3D PDF found at 10.17605/OSF.IO/AT528. Raw reconstructed slices found at 10.17602/M2/M380652.

such as iron pyrite, a high sensitivity to hydrogenous material, and thus to residual organic remains, (*Gee et al., 2019*; *Gee, Bevitt & Reisz, 2019*; *Na et al., 2021*; *Smith et al., 2021*; *Bazanna et al., 2021*), and to increasing availability of high-quality neutron imaging

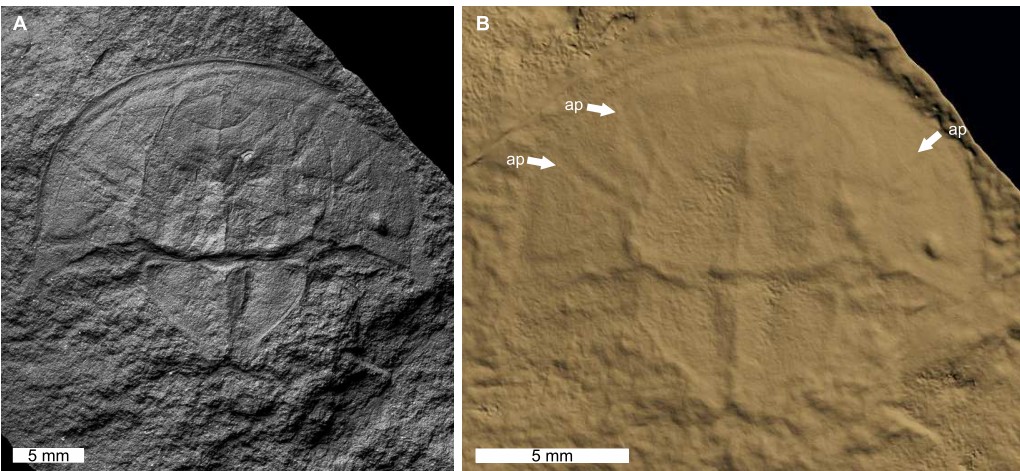

**Figure 2** *Dubbolimulus peetae* **from the Napperby Formation (Middle Triassic, Anisian). MMF 27693, holotype.** (A) Specimen under LED light. (B) 3D reconstruction of specimen showing appendage impressions (white arrows), see Fig. S2. Abbreviation: ap, appendage impression. Image credit: (A) David Barnes. Image in (A) reproduced from *Bicknell & Pates (2020)* under a CC BY 4.0 license. 3D PDF found at 10.17605/OSF.IO/AT528. Raw reconstructed slices found at 10.17602/M2/M396665.

facilities at nuclear research reactors and spallation neutron sources around the world (see https://www.isnr.de/index.php/facilities/user-facilities). Finally, techniques that can more readily distinguish areas with very small differences in radiopacity, such as phase-contrast enhanced imaging, show promise for more detailed examination of muscles and other internal structures in suitably well-preserved specimens. Any, or all of these approaches could be applied to the study of specimens of *Mesolimulus walchi* (*Desmarest, 1822*) from the Nusplingen Lithographic Limestone (Upper Jurassic, Kimmeridgian), Germany, that have muscle traces preserved under the prosoma (*Briggs et al., 2005*). Muscle traces have also been described from specimens of *Euproops danae* from the Upper Pennsylvanian (Virgilian) Lawrence Formation, Kansas (*Feldman et al., 1993*; *Babcock & Merriam, 2000*; *Bicknell et al., 2022b*). Further examination of the Lawrence Formation specimens would determine if the muscles exhibit moldic preservation—as is common for Mazon Creek fossils (*Clements, Purnell & Gabbott, 2019*; *Bicknell et al., 2021c*)—or if there are additional, unexpressed anatomical features. The collection of novel soft anatomy from these and other fossil xiphosurids are vitally important in presenting and revising hypotheses regarding homology with extant xiphosurids (*sensu Briggs et al., 2005*; *Bicknell et al., 2022b*) and resolving conflicts between phylogenetic hypotheses (*e.g., Ballesteros & Sharma, 2019*; *Bicknell, Lustri & Brougham, 2019*; *Bicknell, Naugolnykh & Brougham, 2020*; *Lamsdell, 2020*). More broadly, this same approach can be applied to the as-of-yet unnamed xiphosuran specimens from the Fezouata Shale *Konservat-Lagerstätte* (Lower Ordovician, Morocco; *Van Roy et al., 2010*), as previous micro-CT imagery of material from this deposits has yielded useful results and allowed for specimens to be differentiated in 3D (*Kouraiss et al., 2019*).

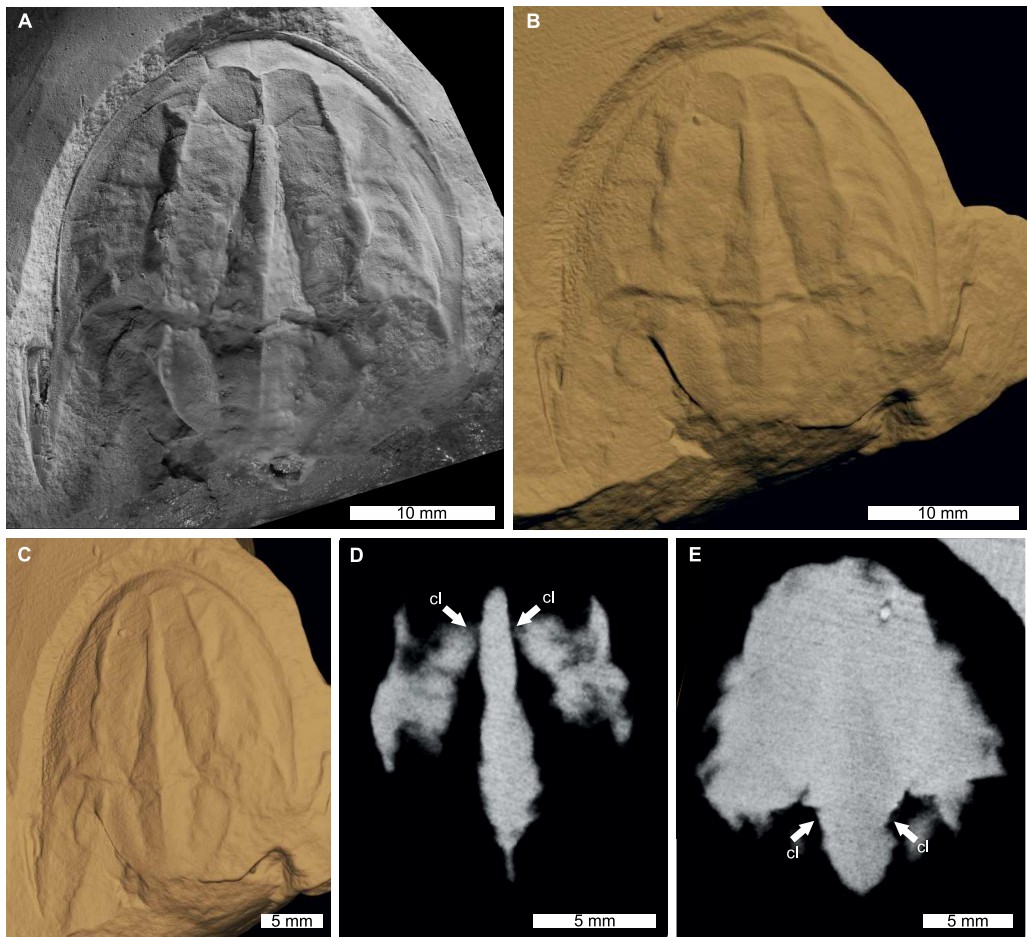

**Figure 3** *Tasmaniolimulus patersoni* from the Jackey Shale (Early Triassic, Induan). UTGD 123979, holotype. (A) Specimen under LED light. (B, C) 3D reconstruction of specimen, see Fig. S3. (B) Dorsal view. (C) Oblique view. (D, E) X-ray tomographic slices showing pronounced cardiac lobe (white arrows). (A) Coated in ammonium chloride sublimate and image converted to greyscale. Abbreviation: cl, cardiac lobe. Image credit: (A) Russell Bicknell. 3D PDF found at 10.17605/OSF.IO/AT528. Raw reconstructed slices found at 10.17602/M2/M396670.

Three-dimensional reconstructions are increasingly used in computational fluid dynamics (CFD) to study the hydrodynamic properties of extinct aquatic taxa (*Rahman et al., 2015a*; *Darroch et al., 2017*; *Rahman, 2017*; *Gibson et al., 2019*; *Ferrón et al., 2020*; *Hebdon, Ritterbush & Choi, 2020*; *Gibson et al., 2021*; *Song et al., 2021*). The majority of CFD studies have focused on enigmatic Ediacaran taxa (*Rahman et al., 2015a*; *Rahman, 2017*; *Gibson et al., 2019*), echinoderms (*Rahman et al., 2015b*; *Rahman et al., 2020*; *Waters et al., 2017*), ammonoids (*Hebdon, Ritterbush & Choi, 2020*), and vertebrate groups (*Dec, 2019*; *Troelsen et al., 2019*; *Ferrón et al., 2020*; *Ferrón et al., 2021*). While fossil arthropods have received comparatively less attention than the aforementioned groups (*e.g.*, *Pates et al., 2021*; *Song et al., 2021*), CFD studies have modelled lift and drag experienced by modern xiphosurids (*Bicknell & Pates, 2019*; *Davis, Hoover & Miller, 2019*). Extending

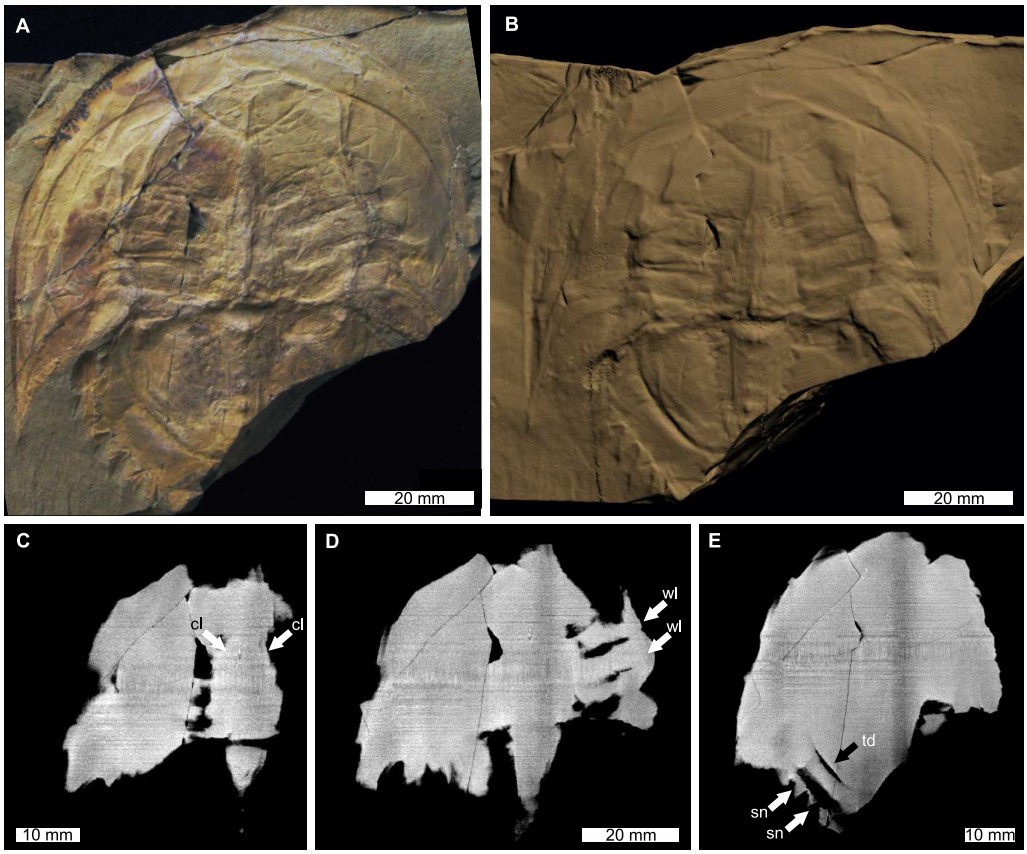

**Figure 4** *Victalimulus mcqueeni* from the Koonwarra Fossil Bed (Early Cretaceous, Aptian). **NMV P22410B, holotype.** (A) Specimen under LED light. (B) 3D reconstruction of specimen, see Fig. S4. (C) X-ray tomographic slice showing cardiac lobe (white arrows). (D) X-ray tomographic slice showing walking leg impressions (white arrows). (E) X-ray tomographic slice showing fixed spines and moveable spine notches (white arrows) and thoracetronic doublure (black arrow). Abbreviations: cl, cardiac lobe; sn, spine notches; td, thoracetronic doublure; wl, walking leg impression. Image credit: (A) Frank Holmes. Image in (A) reproduced from *Bicknell & Pates (2020)* under a CC BY 4.0 license. 3D PDF found at 10.17605/OSF.IO/AT528. Raw reconstructed slices found at 10.17602/M2/M392556.

CFD studies to fossil xiphosurids will facilitate comparative studies of the hydrodynamic properties of the carapaces of extinct members of the clade, in addition to elucidating the effects of bizarre morphologies, such as the hypertrophied genal spines, on fluid flow. Such spines have been hypothesised to represent an adaptation to movement through unidirectional fluid flow in primarily freshwater or marginal marine environments (*Lamsdell, 2016*; *Lamsdell, 2021*; *Bicknell & Pates, 2019*; *Bicknell & Shcherbakov, 2021*; *Bicknell et al., 2022a*); CFD provides the most compelling method for evaluating the likelihood of this hypothesis. Due to compression of the fossils (consider *Dubbolimulus peetae*) CFD models of compressed xiphosurids would need to be retro-deformed, likely using modern forms as a proxy for inflation, to account for taphonomic alteration. However, there are specimens, such as *Crenatolimulus paluxyensis Feldmann et al., 2011*

and *Tachypleus decheni* (*Zincken, 1862*), that have maintained their three-dimensionality (*Bicknell et al., 2021a*). Such specimens may be ideal for scanning and immediate CFD analysis.

Palaeontological and biological collections house a wealth of specimens with academic and historic value. Digitisation of holotype specimens is a salient direction for recording and transferring fundamental anatomical information. These records are traditionally conducted by taking photographs or making line drawings. However, two-dimensional data and views cannot (by definition) display all characteristics needed for modern taxonomic and phylogenetic studies (*Mathys et al., 2015*; *Bicknell et al., 2018a*). As such, researchers often need to visit collections to examine specimens in person. This process can be prohibitive for logistic, cost, and policy reasons, to name a few. This complication can be circumvented by producing scans of taxonomically important and unique specimens. Such data is becoming a means of transferring important anatomical data to researchers across the globe and provide interested individuals with another medium with which to examine unique material (*Hühne, 2018*; *Shi, Westeen & Rabosky, 2018*; *Kouraiss et al., 2019*).

## CONCLUSION

Reconsidering the four Australian xiphosurids here, we have highlighted the rise of Austrolimulidae in the Gondwanan record began just after the end-Permian extinction. This timing also suggests that, globally, the development of hypertrophied spines within non-belinurid xiphosurids began after the end-Permian. We demonstrate that limited novel anatomical data were obtained for *Austrolimulus fletcheri*, *Dubbolimulus peetae*, *Tasmaniolimulus patersoni*, and *Victalimulus mcqueeni* using SRXT, reflecting the preservation of these fossils in sand- and siltstones. Future directions include examining similar fossils with NCT, an additional method that achieves an alternative and complementary contrast to X-ray CT, and may resolve features that conventional lab-based- and synchrotron X-rays are unable to reveal. Future applications of these scan data include informing reconstructions needed for computational fluid dynamic analyses; a direction that may uncover the morpho-functional use of overdeveloped spines common to Australian xiphosurids.

**Institutional acronyms**

| | |
|---|---|
| **AM F** | Australian Museum, Sydney, New South Wales, Australia. |
| **MMF** | Geological Survey of New South Wales, Londonderry, New South Wales, Australia. |
| **NMV P** | Museums Victoria, Carlton, Victoria, Australia. |
| **UTGD** | Geology Department, University of Tasmania, Tasmania, Australia. |

## ACKNOWLEDGEMENTS

We thank Isabella von Lichtan, Matthew McCurry, Rolf Schmidt, and Yong-Yi Zhen for access to the scanned specimens, and Anton Maksimenko for beamline assistance at the Australian Synchrotron. We thank David Barnes, Joshua White, and Frank Holmes

for images. Finally, we thank Carolin Haug and Peter Van Roy for their reviews, and Brandon Hedrick for editorial assistance, both of which improved the scope and focus of the manuscript.

### Funding

This research was supported by funding from a UNE Postdoctoral Research Fellowship (to Russell D.C. Bicknell and Tom Brougham), and an ANSTO research grant (AS1/IMBL/15769 to Russell D.C. Bicknell and Tom Brougham). The funders had no role in study design, data collection and analysis, decision to publish, or preparation of the manuscript.

### Grant Disclosures

The following grant information was disclosed by the authors:
UNE Postdoctoral Research Fellowship.
ANSTO research grant: AS1/IMBL/15769.

### Competing Interests

The authors declare there are no competing interests.

### Author Contributions

- Russell D.C. Bicknell conceived and designed the experiments, performed the experiments, analyzed the data, prepared figures and/or tables, authored or reviewed drafts of the paper, and approved the final draft.
- Patrick M. Smith analyzed the data, prepared figures and/or tables, authored or reviewed drafts of the paper, presented the geological context for the specimens, and approved the final draft.
- Tom Brougham conceived and designed the experiments, performed the experiments, authored or reviewed drafts of the paper, and approved the final draft.
- Joseph J. Bevitt conceived and designed the experiments, performed the experiments, analyzed the data, authored or reviewed drafts of the paper, was central in using the synchrotron beam, and approved the final draft.

### Data Availability

The specimens are located in the following locations:

- *Austrolimulus fletcheri* AM F38275, Australian Museum.
- *Dubbolimulus peetae* MMF 27693, Geological Survey of New South Wales.
- *Victalimulus mcqueeni* NMV P22410B, Museums Victoria.
- *Tasmaniolimulus patersoni* UTGD 123979, Geology Department, University of Tasmania.

The scans are available at MorphoSource:

*Austrolimulus fletcheri*, AM F38275: Raw reconstructed slices, 10.17602/M2/M380652.

*Dubbolimulus peetae*, MMF 27693: Raw reconstructed slices, 10.17602/M2/M396665.

*Tasmaniolimulus patersoni*, UTGD 123979: Raw reconstructed slices, 10.17602/M2/M396670.

*Victalimulus mcqueeni*, NMV P22410B: Raw reconstructed slices, 10.17602/M2/M392556.

The supplementary figures are available at OSF: Bicknell, Russell D. 2022. ''An Earliest Triassic Age for Tasmaniolimulus and Comments on Synchrotron Tomography of Gondwanan Horseshoe Crabs.'' OSF. March 18. doi: 10.17605/OSF.IO/AT528.

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

# PeerJ

**Göpel T, Wirkner CS. 2015.** An ancient complexity? Evolutionary morphology of the circulatory system in Xiphosura. *Zoology* **118(4)**:221–238 DOI 10.1016/j.zool.2014.12.004.

**Gureyev TE, Nesterets Y, Ternovski D, Thompson D, Wilkins SW, Stevenson AW, Sakellariou A, Taylor JA. 2011.** Toolbox for advanced X-ray image processing. *Advances in Computational Methods for X-ray Optics II* **8141**:81410B.

**Haszprunar G, Speimann E, Hawe A, Heß M. 2011.** Interactive 3D anatomy and affinities of the Hyalogyrinidae, basal Heterobranchia (Gastropoda) with a rhipidoglossate radula. *Organisms Diversity & Evolution* **11(3)**:201–236 DOI 10.1007/s13127-011-0048-0.

**Haug C. 2020.** The evolution of feeding within Euchelicerata: data from the fossil groups Eurypterida and Trigonotarbida illustrate possible evolutionary pathways. *PeerJ* **8**:e9696 DOI 10.7717/peerj.9696.

**Haug C, Haug JT. 2020.** Untangling the Gordian knot–further resolving the super-species complex of 300-million-year-old xiphosurids by reconstructing their ontogeny. *Development Genes and Evolution* **230(1)**:13–26 DOI 10.1007/s00427-020-00648-7.

**Haug C, Haug JT, Waloszek D, Maas A, Frattigiani R, Liebau S. 2009.** New methods to document fossils from lithographic limestones of southern Germany and Lebanon. *Palaeontologia Electronica* **12(3)**:12.

**Haug JT, Martin JW, Haug C. 2015.** A 150-million-year-old crab larva and its implications for the early rise of brachyuran crabs. *Nature Communications* **6(1)**:1–6.

**Haug JT, Müller P, Haug C. 2019.** A 100-million-year old predator: a fossil neuropteran larva with unusually elongated mouthparts. *Zoological Letters* **5(1)**:1–14 DOI 10.1186/s40851-018-0113-z.

**Haug C, Rötzer MAIN. 2018.** The ontogeny of *Limulus polyphemus* (Xiphosura s. str., Euchelicerata) revised: looking under the skin. *Development Genes and Evolution* **228(1)**:49–61 DOI 10.1007/s00427-018-0603-1.

**Haug C, Van Roy P, Leipner A, Funch P, Rudkin DM, Schöllmann L, Haug JT. 2012.** A holomorph approach to xiphosuran evolution—a case study on the ontogeny of *Euproops*. *Development Genes and Evolution* **222(5)**:253–268 DOI 10.1007/s00427-012-0407-7.

**Hebdon N, Ritterbush KA, Choi Y. 2020.** Computational fluid dynamics modeling of fossil ammonoid shells. *Palaeontologia Electronica* **23(1)**:a21.

**Hegna TA, Martin MJ, Darroch SAF. 2017.** Pyritized *in situ* trilobite eggs from the Ordovician of New York (Lorraine Group): implications for trilobite reproductive biology. *Geology* **45(3)**:199–202.

**Helby R. 1973.** Review of Late Permian and Triassic palynology of New South Wales. *Special Publications of the Geological Society of Australia* **4**:141–155.

**Helby R, Morgan R, Partridge AD. 1987.** A palynological zonation of the Australian Mesozoic. *Memoir of the Association of Australasian Palaeontologists* **4**:1–94.

**Henderickx H, Tafforeau P, Soriano C. 2012.** Phase contrast synchrotron microtomography reveals the morphology of a partially visible new *Pseudogarypus* in Baltic amber (Pseudoscorpiones: Pseudogarypidae). *Palaeontologia Electronica* **15(2)**:1–11.

**Herbert C. 1980.** Depostional development of the Sydney Basin. In: Herbert C, Helby R, eds. *A guide to the Sydney Basin*. 26. Sydney: Department of Mineral Resources Geological Survey of NSW, 11–25.

**Herbert C. 1983.** Sydney Basin stratigraphy. In: Herbert C, ed. *Geology of the Sydney Basin 1:100,000 sheet 9130.* Sydney: New South Wales Department of Natural Resources.

**Herbert G. 1997.** Sequence stratigraphic analysis of early and middle Triassic alluvial and estuarine fades in the Sydney Basin, Australia. *Australian Journal of Earth Sciences* **44(1)**:125–143 DOI 10.1080/08120099708728299.

**Herrera F, Shi G, Mays C, Ichinnorov N, Takahashi M, Bevitt JJ, Herendeen PS, Crane PR. 2020.** Reconstructing *Krassilovia mongolica* supports recognition of a new and unusual group of Mesozoic conifers. *PLoS ONE* **15(1)**:e0226779 DOI 10.1371/journal.pone.0226779.

**Hita Garcia F, Fischer G, Liu C, Audisio TL, Alpert GD, Fisher BL, Economo EP. 2017.** X-ray microtomography for ant taxonomy: An exploration and case study with two new *Terataner* (Hymenoptera, Formicidae, Myrmicinae) species from Madagascar. *PLoS ONE* **12(3)**:e0172641 DOI 10.1371/journal.pone.0172641.

**Holmes WBK. 1982.** The Middle Triassic flora from Benolong, near Dubbo, central-western New South Wales. *Alcheringa* **6(1)**:1–33 DOI 10.1080/03115518208565416.

**Hühne C. 2018.** Scientific methods of geological and paleontological collections and trends in paleontological investigation and research. In: Beck LA, Joger U, eds. *Paleontological Collections of Germany, Austria and Switzerland: the history of life of fossil organisms at museums and universities.* Cham: Springer International Publishing, 15–22.

**Jell PA, Roberts J. 1986.** Plants and invertebrates from the Lower Cretaceous Koonwarra Fossil Bed, South Gippsland, Victoria. *Memoirs of the Association of Australasian Palaeontologists* **3**:1–77.

**Klein N. 2012.** Postcranial morphology and growth of the pachypleurosaur *Anarosaurus heterodontus* (Sauropterygia) from the Lower Muschelkalk of Winterswijk, The Netherlands. *Paläontologische Zeitschrift* **86(4)**:389–408 DOI 10.1007/s12542-012-0137-1.

**Klompmaker AA, Fraaije RHB. 2011.** The oldest (Middle Triassic, Anisian) lobsters from the Netherlands: taxonomy, taphonomy, paleoenvironment, and paleoecology. *Palaeontologia Electronica* **14(1)**:1–16.

**Klug C, Hagdorn H, Montenari M. 2005.** Phosphatized soft-tissue in Triassic bivalves. *Palaeontology* **48(4)**:833–852 DOI 10.1111/j.1475-4983.2005.00485.x.

**Korasidis VA, Wagstaff BE. 2020.** The rise of flowering plants in the high southern latitudes of Australia. *Review of Palaeobotany and Palynology* **272**:104126 DOI 10.1016/j.revpalbo.2019.104126.

**Kouraiss K, Hariri KEl, Albani AEl, Azizi A, Mazurier A, Lefebvre B. 2019.** Digitization of fossils from the Fezouata Biota (Lower Ordovician, Morocco): Evaluating computed tomography and photogrammetry in collection enhancement. *Geoheritage* **11(4)**:1889–1901 DOI 10.1007/s12371-019-00403-z.

**Kutara K, Une Y, Fujita Y. 2019.** Morphological assessment of horseshoe crabs (*Tachypleus tridentatus*) by using magnetic resonance imaging. *Journal of Zoo and Wildlife Medicine* **50(3)**:742–748 DOI 10.1638/2018-0195.

**Lak M, Néraudeau D, Nel A, Cloetens P, Perrichot V, Tafforeau P. 2008.** Phase contrast X-ray synchrotron imaging: opening access to fossil inclusions in opaque amber. *Microscopy and Microanalysis* **14(3)**:251–259 DOI 10.1017/S1431927608080264.

**Lamsdell JC. 2016.** Horseshoe crab phylogeny and independent colonizations of fresh water: ecological invasion as a driver for morphological innovation. *Palaeontology* **59(2)**:181–194 DOI 10.1111/pala.12220.

**Lamsdell JC. 2020.** The phylogeny and systematics of Xiphosura. *PeerJ* **8**:e10431 DOI 10.7717/peerj.10431.

**Lamsdell JC. 2021.** A new method for quantifying heterochrony in evolutionary lineages. *Paleobiology* **47(2)**:363–384 DOI 10.1017/pab.2020.17.

**Landschoff J, Komai T, Plessis ADu, Gouws G, Griffiths CL. 2018.** MicroCT imaging applied to description of a new species of *Pagurus* Fabricius, 1775 (Crustacea: Decapoda: Anomura: Paguridae), with selection of three-dimensional type data. *PLoS ONE* **13(9)**:e0203107 DOI 10.1371/journal.pone.0203107.

**Lange W. 1923.** Über neue Fossilfunde aus der Trias von Göttingen. *Zeitschrift der deutschen geologischen Gesellschaft* **74**:162–168.

**Latreille PA. 1802.** *Histoire naturelle, générale et particulière, des crustacés et des insectes.* Paris: Dufart.

**Laurie JR, Bodorkos S, Nicoll RS, Crowley JL, Mantle DJ, Mory AJ, Wood GR, Backhouse J, Holmes EK, Smith TE. 2016.** Calibrating the middle and late Permian palynostratigraphy of Australia to the geologic time-scale via U–Pb zircon CA-IDTIMS dating. *Australian Journal of Earth Sciences* **63(6)**:701–730 DOI 10.1080/08120099.2016.1233456.

**Leach WE. 1819.** Entomostraca. In: Levrault F, ed. *Dictionaire des science naturelles.* Paris: Levrault and Schoell, 524–543.

**Lerner AJ, Lucas SG, Lockley M. 2017.** First fossil horseshoe crab (Xiphosurida) from the Triassic of North America. *Neues Jahrbuch für Geologie und Paläontologie-Abhandlungen* **286(3)**:289–302 DOI 10.1127/njgpa/2017/0702.

**Lindsay NM. 1982.** The Burial History of the Strzelecki Group Sandstones, S.E. Australia: a petrographic and fission track study. Unpublished M.Sc. thesis, University of Melbourne, Melbourne.

**Linnaeus C. 1758.** *Systema naturæper regna tria naturæ, secundum classes, ordines, genera, species, cum characteribus, differentiis, synonymis, locis.* Holmiae: Laurentius Salvius.

**Liu T, Duan B, Zhang H, Cheng G, Liu J, Dong X-P, Waloszek D, Maas A. 2019.** Soft-tissue anatomy of an Orsten-type phosphatocopid crustacean from the Cambrian

Furongian of China revealed by synchrotron radiation X-ray tomographic microscopy. *Neues Jahrbuch für Geologie und Paläontologie-Abhandlungen* **294**:263–274 DOI 10.1127/njgpa/2019/0858.

**Liu Y, Melzer RR, Haug JT, Haug C, Briggs DE, Hörnig MK, He Y-Y, Hou X-G. 2016.** Three-dimensionally preserved minute larva of a great-appendage arthropod from the early Cambrian Chengjiang biota. *Proceedings of the National Academy of Sciences* **113(20)**:5542–5546 DOI 10.1073/pnas.1522899113.

**Liu Y, Ortega-Hernández J, Chen H, Mai H, Zhai D, Hou X. 2020.** Computed tomography sheds new light on the affinities of the enigmatic euarthropod *Jianshania furcatus* from the early Cambrian Chengjiang biota. *BMC Evolutionary Biology* **20**:1 DOI 10.1186/s12862-019-1549-2.

**Liu W, Rühr PT, Wesener T. 2017.** A look with μCT technology into a treasure trove of fossils: The first two fossils of the millipede order Siphoniulida discovered in Cretaceous Burmese amber (Myriapoda, Diplopoda). *Cretaceous Research* **74**:100–108 DOI 10.1016/j.cretres.2017.01.009.

**MacDougall MJ, Seeger R, Gee B, Ponstein J, Jansen M, Scott D, Bevitt JJ, Reisz RR, Fröbisch J. 2021.** Revised description of the Early Permian recumbirostran microsaur *Nannaroter mckinziei* based on new fossil material and computed tomographic data. *Fronteirs in Ecology and Evolution* **9**:739316 DOI 10.3389/fevo.2021.739316.

**Marcondes Machado F, Passos FD, Giribet G. 2019.** The use of micro-computed tomography as a minimally invasive tool for anatomical study of bivalves (Mollusca: Bivalvia). *Zoological Journal of the Linnean Society* **186(1)**:46–75 DOI 10.1093/zoolinnean/zly054.

**Mathys A, Brecko J, Vandenspiegel D, Cammaert L, Semal P. 2015.** Bringing collections to the digital era three examples of integrated high resolution digitisation projects. In: *Proceedings of the Digital Heritage 2015, Spain*. Granada: Digital Heritage, 155–158.

**Mayr G, De Pietri VL, Love L, Mannering AA, Bevitt JJ, Scofield RP. 2020.** First complete wing of a stem group sphenisciform from the Paleocene of New Zealand sheds light on the evolution of the penguin flipper. *Diversity* **12(2)**:46 DOI 10.3390/d12020046.

**Mays C, Vajda V, Frank TD, Fielding CR, Nicoll RS, Tevyaw AP, McLoughlin S. 2020.** Refined Permian–Triassic floristic timeline reveals early collapse and delayed recovery of south polar terrestrial ecosystems. *Geological Society of America Bulletin* **132(7–8)**:1489–1513 DOI 10.1130/B35355.1.

**Meischner K-D. 1962.** Neue Funde von *Psammolimulus gottingensis* (Merostomata, Xiphosura) aus dem Mittleren Buntsandstein von Göttingen. *Paläontologische Zeitschrift* **36(1)**:185–193 DOI 10.1007/BF02987900.

**Metcalfe I, Crowley JL, Nicoll RS, Schmitz M. 2015.** High-precision U-Pb CA-TIMS calibration of Middle Permian to Lower Triassic sequences, mass extinction and

extreme climate-change in eastern Australian Gondwana. *Gondwana Research* **28(1)**:61–81 DOI 10.1016/j.gr.2014.09.002.

**Metscher BD. 2009.** MicroCT for comparative morphology: simple staining methods allow high-contrast 3D imaging of diverse non-mineralized animal tissues. *BMC Physiology* **9(1)**:11 DOI 10.1186/1472-6793-9-11.

**Moreau J-D, Cloetens P, Gomez B, Daviero-Gomez V, Néraudeau D, Lafford TA, Tafforeau P. 2014.** Multiscale 3D virtual dissections of 100-million-year-old flowers using X-ray synchrotron micro-and nanotomography. *Microscopy and Microanalysis* **20(1)**:305–312 DOI 10.1017/S1431927613014025.

**Motchurova-Dekova N, Harper DAT. 2010.** Synchrotron radiation X-ray tomographic microscopy (SRXTM) of brachiopod shell interiors for taxonomy: preliminary report. *Annales Géologiques de la Péninsule Balkanique* **71**:109–117.

**Na Y, Sun C, Wang H, Huang T, Bevitt J, Li Y, Li T, Zhao Y, Li N. 2021.** Application of neutron tomography in studying new material of *Ixostrobus* Raciborski from the Middle Jurassic of Inner Mongolia, China. *Geological Journal* **56(9)**:4618–4626 DOI 10.1002/gj.4196.

**Parapar J, Candás M, Cunha-Veira X, Moreira J. 2017.** Exploring annelid anatomy using micro-computed tomography: A taxonomic approach. *Zoologischer Anzeiger* **270**:19–42 DOI 10.1016/j.jcz.2017.09.001.

**Pardo JD, Anderson JS. 2016.** Cranial morphology of the Carboniferous-Permian tetrapod *Brachydectes newberryi* (Lepospondyli, Lysorophia): new data from µCT. *PLoS ONE* **11(8)**:e0161823 DOI 10.1371/journal.pone.0161823.

**Pates S, Daley AC, Legg DA, Rahman IA. 2021.** Vertically migrating *Isoxys* and the early Cambrian biological pump. *Proceedings of the Royal Society of London B* **288(1953)**:20210464.

**Perrichot V, Marion L, Neraudeau D, Vullo R, Tafforeau P. 2008.** The early evolution of feathers: fossil evidence from Cretaceous amber of France. *Proceedings of the Royal Society of London B* **275(1639)**:1197–1202.

**Pickett JW. 1984.** A new freshwater limuloid from the middle Triassic of New South Wales. *Palaeontology* **27(3)**:609–621.

**Pike GP. 1973.** Quamby, Tasmania. Geological Atlas 1 Mile Series Explanatory Report, Sheet 46 (8219N). Tasmanian Department of Mines, Hobart.

**Pohl H, Wipfler B, Grimaldi D, Beckmann F, Beutel R. 2010.** Reconstructing the anatomy of the 42-million-year-old fossil *Mengea tertiaria* (Insecta, Strepsiptera). *Naturwissenschaften* **97(9)**:855–859 DOI 10.1007/s00114-010-0703-x.

**Poropat SF, Martin SK, Tosolini A-MP, Wagstaff BE, Bean LB, Kear BP, Vickers-Rich P, Rich TH. 2018.** Early Cretaceous polar biotas of Victoria, southeastern Australia—an overview of research to date. *Alcheringa* **42(2)**:157–229 DOI 10.1080/03115518.2018.1453085.

**Price PL. 1997.** Permian to Jurassic palynostratigraphic nomenclature of the Bowen and Surat Basin. In: Green PM, ed. *The Surat and Bowen Basins, South-East Queensland.*

*Queensland minerals and energy review series*, Brisbane: Queensland Department of Mines and Energy, 137–178.

**Rahman IA. 2017.** Computational fluid dynamics as a tool for testing functional and ecological hypotheses in fossil taxa. *Palaeontology* **60(4)**:451–459 DOI 10.1111/pala.12295.

**Rahman IA, Darroch SAF, Racicot RA, Laflamme M. 2015a.** Suspension feeding in the enigmatic Ediacaran organism *Tribrachidium* demonstrates complexity of Neoproterozoic ecosystems. *Science Advances* **1(10)**:e1500800 DOI 10.1126/sciadv.1500800.

**Rahman IA, O'Shea J, Lautenschlager S, Zamora S. 2020.** Potential evolutionary trade-off between feeding and stability in Cambrian cinctan echinoderms. *Palaeontology* **63(5)**:689–701 DOI 10.1111/pala.12495.

**Rahman IA, Zamora S, Falkingham PL, Phillips JC. 2015b.** Cambrian cinctan echinoderms shed light on feeding in the ancestral deuterostome. *Proceedings of the Royal Society of London B* **282(1818)**:20151964.

**Raymond CA, Bevitt JJ, Tristant Y, Power RK, Lanati AW, Davey CJ, Magnussen JS, Clark SM. 2019.** Recycled blessings: An investigative case study of a rewrapped egyptian votive mummy using novel and established 3D imaging techniques. *Archaeometry* **61(5)**:1160–1174 DOI 10.1111/arcm.12477.

**Reid M, Bordy EM, Taylor WL, Le Roux SG, Du Plessis A. 2019.** A micro X-ray computed tomography dataset of fossil echinoderms in an ancient obrution bed: a robust method for taphonomic and palaeoecologic analyses. *GigaScience* **8(3)**:giy156 DOI 10.1093/gigascience/giy156.

**Retallack GJ. 1977.** Reconstructing Triassic vegetation of eastern Australasia: a new approach for the biostratigraphy of Gondwanaland. *Alcheringa* **1(3)**:247–277 DOI 10.1080/03115517708527763.

**Retallack GJ. 1980.** Late Carboniferous to Middle Triassic megafossil floras from the Sydney Basin. In: Herbert C, Helby R, eds. *A guide to the Sydney Basin*. 26. Sydney: Department of Mineral Resources Geological Survey of NSW, 385–430.

**Riedel A, Dos Santos Rolo T, Cecilia A, Van de Kamp T. 2012.** Sayrevilleinae Legalov, a newly recognised subfamily of fossil weevils (Coleoptera, Curculionoidea, Attelabidae) and the use of synchrotron microtomography to examine inclusions in amber. *Zoological Journal of the Linnean Society* **165(4)**:773–794 DOI 10.1111/j.1096-3642.2012.00825.x.

**Riek EF. 1955.** A new xiphosuran from the Triassic sediments at Brookvale, New South Wales. *Records of the Australian Museum* **23(5)**:281–282 DOI 10.3853/j.0067-1975.23.1955.637.

**Riek EF, Gill ED. 1971.** A new xiphosuran genus from Lower Cretaceous freshwater sediments at Koonwarra, Victoria, Australia. *Palaeontology* **14(2)**:206–210.

**Rust BR, Jones BG. 1987.** The Hawkesbury Sandstone south of Sydney, Australia; Triassic analogue for the deposit of a large, braided river. *Journal of Sedimentary Research* **57(2)**:222–233.

**Sander PM, Wintrich T, Schwermann AH, Kindlimann R. 2016.** Die paläontologische Grabung in der Rhät-Lias-Tongrube der Fa. Lücking bei Warburg-Bonenburg (Kr. Höxter) im Frühjahr 2015. *Geologie und Paläontologie in Westfalen* **88**:11–37.

**Schimpf L, Isaak S, Hauschke N, Gossel W. 2017.** Computer-generated 3D models and digital storage for use in palaeontological collections, tested for xiphosurans of Eocene age from Saxony-Anhalt, Germany. *Hallesches Jahrbuch für Geowissenschaften* **40**:1–16.

**Schwarzhans W, Beckett HT, Schein JD, Friedman M. 2018.** Computed tomography scanning as a tool for linking the skeletal and otolith-based fossil records of teleost fishes. *Palaeontology* **61(4)**:511–541 DOI 10.1111/pala.12349.

**Seegets-Villiers DE, Wagstaff BE. 2016.** Morphological variation of stratigraphically important species in the genus *Pilosisporites* Delcourt & Sprumont, 1955 in the Gippsland Basin, southeastern Australia. *Memoirs of Museum Victoria* **74**:81–91 DOI 10.24199/j.mmv.2016.74.08.

**Shi JJ, Westeen EP, Rabosky DL. 2018.** Digitizing extant bat diversity: An open-access repository of 3D μCT-scanned skulls for research and education. *PLoS ONE* **13(9)**:e0203022 DOI 10.1371/journal.pone.0203022.

**Smith HE, Bevitt JJ, Zaim J, Rizal Y, Puspaningrum MR, Trihascaryo A, Price GJ, Webb GE, Louys J. 2021.** High-resolution high-throughput thermal neutron tomographic imaging of fossiliferous cave breccias from Sumatra. *Scientific Reports* **11(1)**:1–16 DOI 10.1038/s41598-020-79139-8.

**Snyder AJ, LeBlanc ARH, Jun C, Bevitt JJ, Reisz RR. 2020.** Thecodont tooth attachment and replacement in bolosaurid parareptiles. *PeerJ* **8**:e9168 DOI 10.7717/peerj.9168.

**Song H, Song H, Rahman IA, Chu D. 2021.** Computational fluid dynamics confirms drag reduction associated with trilobite queuing behaviour. *Palaeontology* **64(5)**:597–608 DOI 10.1111/pala.12562.

**Stillwell JD, Buckeridge J St JS, Bevitt JJ, Zahra D. 2020.** Fossil barnacles from the Antarctic Peninsula: refining ways of exploring the nature of rare and/or delicate specimens employing X-ray Computer Tomography (CT). *Journal of Palaeontology* **94(6)**:1076–1081 DOI 10.1017/jpa.2020.33.

**Størmer L. 1952.** Phylogeny and taxonomy of fossil horseshoe crabs. *Journal of Paleontology* **26(4)**:630–640.

**Sutton MD. 2008.** Tomographic techniques for the study of exceptionally preserved fossils. *Proceedings of the Royal Society of London B* **275(1643)**:1587–1593.

**Tadros NZ. 1993.** Gunnedah Basin, New South Wales. *Geological Survey of New South Wales, Memoir (Geology)* **12**:1–649.

**Tafforeau P, Boistel R, Boller E, Bravin A, Brunet M, Chaimanee Y, Cloetens P, Feist M, Hoszowska J, Jaeger J-J. 2006.** Applications of X-ray synchrotron microtomography for non-destructive 3D studies of paleontological specimens. *Applied Physics A* **83(2)**:195–202 DOI 10.1007/s00339-006-3507-2.

**Troelsen PV, Wilkinson DM, Seddighi M, Allanson DR, Falkingham PL. 2019.** Functional morphology and hydrodynamics of plesiosaur necks: does size matter? *Journal of Vertebrate Paleontology* **39(2)**:e1594850 DOI 10.1080/02724634.2019.1594850.

**Van Roy P, Orr PJ, Botting JP, Muir LA, Vinther J, Lefebvre B, Hariri KEl, Briggs DEG. 2010.** Ordovician faunas of Burgess Shale type. *Nature* **465(7295)**:215–218 DOI 10.1038/nature09038.

**Vía L, De Villata FJ, Esteban Cerdá M. 1977.** Paleontología y Paleoecología de los yacimientos fosilíferos del Muschelkalk superior entre Alcover y Mont-Ral (Montañas de Prades, provincia de Tarragona). *Journal of Iberian Geology* **4**:247–258.

**Wagstaff BE, Gallagher SJ, Hall WM, Korasidis VA, Rich TH, Seegets-Villiers DE, Vickers-Rich PA. 2020.** Palynological-age determination of Early Cretaceous vertebrate-bearing beds along the south Victorian coast of Australia, with implications for the spore-pollen biostratigraphy of the region. *Alcheringa* **44(3)**:460–474 DOI 10.1080/03115518.2020.1754464.

**Waldman M. 1971.** Fish from the freshwater Lower Cretaceous of Victoria, Australia, with comments on the palaeoenvironment. *Special Papers in Palaeontology* **9**:1–124.

**Waldman M. 1973.** The fossil lake-fauna of Koonwarra, Victoria. *Australian Natural History* **17**:317–321.

**Waldman M. 1984.** The fossil lake-fauna of Koonwarra, Victoria. In: Archer M, Clayton G, eds. *Vertebrate zoogeography & evolution in Australasia (Animals in space and time)*. Sydney: Hesperian Press, 231–233.

**Waters JA, White LE, Sumrall CD, Nguyen BK. 2017.** A new model of respiration in blastoid (Echinodermata) hydrospires based on computational fluid dynamic simulations of virtual 3D models. *Journal of Paleontology* **91(4)**:662–671 DOI 10.1017/jpa.2017.1.

**Webby BD. 1970.** Brookvalichnus, a new trace fossil from the Triassic of the Sydney Basin, Australia. *Geological Journal Special Issue* **3**:527–530.

**Wesener T. 2019.** The oldest pill millipede fossil: a species of the Asiatic pill millipede genus *Hyleoglomeris* in Baltic amber (Diplopoda: Glomerida: Glomeridae). *Zoologischer Anzeiger* **283**:40–45 DOI 10.1016/j.jcz.2019.08.009.

**Willsch M, Friedrich F, Baum D, Jurisch I, Ohl M. 2020.** A comparative description of the mesosomal musculature in Sphecidae and Ampulicidae (Hymenoptera, Apoidea) using 3D techniques. *Deutsche Entomologische Zeitschrift* **67(1)**:51–67 DOI 10.3897/dez.67.49493.

**Xing L, Caldwell MW, Chen R, Nydam RL, Palci A, Simões TR, McKellar RC, Lee MSY, Liu Y, Shi H. 2018.** A mid-Cretaceous embryonic-to-neonate snake in amber from Myanmar. *Science Advances* **4(7)**:eaat5042 DOI 10.1126/sciadv.aat5042.

**Xing L, McKellar RC, Wang M, Bai M, O'Connor JK, Benton MJ, Zhang J, Wang Y, Tseng K, Lockley MG. 2016a.** Mummified precocial bird wings in mid-Cretaceous Burmese amber. *Nature Communications* **7(1)**:1–7.

**Xing L, McKellar RC, Xu X, Li G, Bai M, Persons IVWS, Miyashita T, Benton MJ, Zhang J, Wolfe AP. 2016b.** A feathered dinosaur tail with primitive plumage trapped

in mid-Cretaceous amber. *Current Biology* **26(24)**:3352–3360 DOI 10.1016/j.cub.2016.10.008.

**Young GC, Laurie JR. 1966.** *An Australian phanerozoic timescale.* Melbourne: Oxford University Press.

**Yuen AHL, Kwok DHC, Kim SW. 2019.** Magnetic resonance imaging of the live tri-spine horseshoe crab (*Tachypleus tridentatus*). *Arthropoda Selecta* **28(2)**:247–251 DOI 10.15298/arthsel.28.2.06.

**Zhai D, Edgecombe GD, Bond AD, Mai H, Hou X, Liu Y. 2019a.** Fine-scale appendage structure of the Cambrian trilobitomorph *Naraoia spinosa* and its ontogenetic and ecological implications. *Proceedings of the Royal Society of London B* **286(1916)**:20192371.

**Zhai D, Ortega Hernández J, Wolfe JM, Hou X, Cao C, Liu Y. 2019b.** Three-dimensionally preserved appendages in an early Cambrian stem-group pancrustacean. *Current Biology* **29(1)**:171–177 DOI 10.1016/j.cub.2018.11.060.

**Zincken C. 1862.** *Limulus decheni* aus dem Braunkohlensandstein bei Teuchern. *Zeitschrift für die Gesammten Naturwissenschaften* **19**:329–331.

**Zuber M, Laaß M, Hamann E, Kretschmer S, Hauschke N, Van De Kamp T, Baumbach T, Koenig T. 2017.** Augmented laminography, a correlative 3D imaging method for revealing the inner structure of compressed fossils. *Scientific Reports* **7**:41413 DOI 10.1038/srep41413.