# Peer review of "An earliest Triassic age for Tasmaniolimulus and comments on synchrotron tomography of Gondwanan horseshoe crabs"

_PeerJ, doi:10.7717/peerj.13326_

## Round 0.1 · original submission · Major Revisions

Dear authors,

Thank you for your submission to PeerJ. Following comments from two reviewers, I believe that this manuscript will be publishable in PeerJ after major revisions.

Reviewer 1 notes that the authors should compare their synchrotron data with stereographic images rather than flat images. This would be a valuable addition to the paper given that the goal of the paper is to establish the utility of synchrotron-based microCT in comparison with more conventional methods. Expansion on the explanation of the age of the taxa is also necessary (see reviewer 1 comments).

Reviewer 2 highlights a number of points that deserve additional consideration and explanation in the text. Additionally, the end of the paper discusses the importance of digitization of natural history collections. There is quite a large literature on this topic and I think this part of the paper could be expanded some with additional references. Democratizing natural history collections via digitization is vitally important. I have highlighted a few minor comments below as well from my reading of the manuscript.

When submitting your revision, please include a tracked changes version of the manuscript, a clean version of the manuscript, and a response to reviewer points. Please let me know if you have any questions and I will be happy to answer them. Thank you for your submission!

Best,

Brandon P. Hedrick, Ph.D.


Minor comments:

Line 100: Should this be macro rather than makro?

Line 120: accidental period at the end of this line. Also, can you define ‘remaining components’?

Line 142: Need period between ‘…zone An…’

Line 175: ‘however, it is considered…’. Also that’s a high degree of precision if the collection site is unknown. What are these coordinates based on?

Line 240–255: This paragraph is under the header relating to the ‘age of Tasmaniolimulus’, but discusses anatomy instead. Perhaps expand the header to be ‘Age and Anatomy of…’

Line 255: ‘potential’. Are there any stats on this? Otherwise, I would tone the statement down a bit.

Line 262–267: The digitization of natural history collections is currently a vibrant field and is being widely published on. Add in some citations to these sentences reflecting that recent work.

Line 295–310: How do you think taphonomic compression and distortion would impact CFD analyses on horseshoe crabs? Can you expand on this some here?

Add labels to white arrows in the figures. The structures are not easy to see and I think labels would really improve readability, especially since the arrows refer to different structures in different figures.

·

Basic reporting

In general, the idea of scanning rare fossil material in the synchrotron is good as it might reveal internal structures otherwise not accessible. However, the authors produced 3D surface models (which is fine), which they then compare with “normal” photos of different quality, especially concerning the light settings (see below). They then emphasise that with their method, structures become visible which are not accessible with normal photographic techniques, which I doubt for most cases. In my own experience, especially also with fossil xiphosurans, it is possible to access details of the surface structures with normal photographic equipment, which even allows stereo imaging (see also references mentioned below). Therefore, the authors should not compare their 3D models with normal photographs, but with stereo images of the same specimens.

Additionally, the manuscript has a second focus on the age of one of the species. However, there is no clear explanation in the methods part how the authors came to these conclusions, but a very lengthy geological context section, in which it does not become clear, which results are their own ones and which ones were already known previously (see also details below).

I think that this manuscript could be published after major revision.

Experimental design

Geological context (lines 128-213): This entire section, which is about 3.5 pages long, is not mentioned in the abstract at all. Also, I am not sure what the authors want to tell with this section. Apparently, these data are already mostly known as many references are cited here. Also, this section does not contribute much to the topic of the paper. Therefore, I suggest to delete most of it ans to only point out the aspects important for the context of this study.
The part on the age of Tasmaniolimulus would probably be interesting to keep, but in the Results section and more focussed on which aspects are new, so what the authors have actually done and found out. There is also no explanation in the material and methods section what the authors did to find our about the age of the specimens.

3D reconstructions: I think it would be necessary to compare the virtual 3D reconstructions, which the authors show in their figures, with simple stereo images taken from two different angles. In my experience, these simple images provide very good 3D information (e.g. as red-cyan stereo anaglyphs) and can be produced with simple camera equipment or even a flatbed scanner. Additionally, such simple stereo images do not go through an interpretation process, in contrast to the production of the 3D reconstructions, which the authors did.

Figures: It is stated that “plain light” was used. However, it is clear that in Fig. 1A and 2A (for example, but also in some other images) the lighting was set differently. In Fig. 2A, the light was to a certain extent directed, which is why shadows are visible. Therefore, the appendages are well visible in Fig. 2A (maybe even more clearly than in Fig. 2B). The authors need to rephrase the captions to make clear, which lighting setting was used in the different images, especially as they are pointing out the differences in the methods. They should also provide a statement about the virtual lighting for the renders of the models. If, for example, a ray-tracing algorithms was used, the influence of the lamps can be quite drastic.

General comment: The term horseshoe crabs, though common, is rather unfortunate as these animals are not at all closely related to crabs (which the authors are clearly aware of). Better change to Xiphosurida or whichever other name is preferred.

Validity of the findings

see other comments

Additional comments

24: Though synchrotron CT (and also other CT types) are often termed non-destructive, this is not really the case. The method causes some destruction, though not much, depending on the type of fossils or on the mounting method etc. Please rephrase here and in other places where necessary.

35: Use Euchelicerata instead of Euchelicerate

47: Use Metazoa instead of Animalia, which is rather outdated

63-67: I would simply delete these brackets [ ] as they decrease the readability.

69-70: You wrote “most 3D data collected from fossil xiphosurids have been surface scans”, but cite only one paper, please include some more references here. In general, normal stereo imaging is not uncommon in palaeontological papers and would fit here. Also, I do not want to do citation pushing, but a suitable paper would be this one (especially check fig. 3e):
Haug, C. & Rötzer, M. A. I. N. 2018. The ontogeny of the 300 million year old xiphosuran Euproops danae (Euchelicerata) and implications for resolving the Euproops species complex. Development Genes and Evolution 228, 63–74.

86: Change to “species of Xiphosurida”.

91: Change “taxa” to “species”

92: I recommend to avoid Linnean ranks such as families (rank height is very subjective). Here, it could simply be substituted by ingroups.

98: cm is no SI unit, use mm instead.

129: Use Australian instead of Australia

184: I think it should read “sufficiently”?

226: Use species instead of taxon (also check in other places).

214 ff.: The whole results section is very short. Maybe some more description would make it easier for the reader to follow.

231: As said above, it is not clear how the authors found out about the new age (that means, which methods they used here, that should be in the methods section).

231: Why does the revised age have “important implications for austrolimulid evolution”? The age of fossils does not directly affect the phylogenetic relationships, and oldest fossils do not necessarily have the most ancestral morphology. Please explain what you mean here, the following sentences are not really sufficient here.

252-254: Again, I do not aim at pushing own papers, but as the authors mention that “there are limited examples”, these two show very good examples for ventral preservation (which are also shown as stereo images):
1) Haug, C. & Rötzer, M. A. I. N. 2018. The ontogeny of the 300 million year old xiphosuran Euproops danae (Euchelicerata) and implications for resolving the Euproops species complex. Development Genes and Evolution 228, 63–74.
2) Haug, C., Van Roy, P., Leipner, A., Funch, P., Rudkin, D. M., Schöllmann, L. & Haug, J. T. 2012. A holomorph approach to xiphosuran evolution—a case study on the ontogeny of Euproops. Development Genes and Evolution 222, 253–268.

275: carapace is a tricky term, better use prosomal shield

298-301: But CFD has also been used, e.g., for ammonites: https://palaeo-electronica.org/content/2020/3003-cfd-of-ammonoid-shells
What about CFD studies on large marine crustaceans, are there any?

Caption of Fig. 4: Reference to C is missing, instead F is referred though not in figure, needs to be corrected.

·

Basic reporting

The paper is mostly well-written, with sufficient background and references to the relevant literature provided. There are a number of typos in the text, and in the attached annotated manuscript, I have made corrections and suggested some rephrasing of sentences in sticky notes. The figures are sufficient, of good quality, and backed up by further supplementary images uploaded online.

However, in some cases, from the text, it is not entirely clear what the authors are trying to say. For a detailed discussion of these issues, please see the "Additional comments", and the annotated manuscript.

Experimental design

So far, only one single fossil xiphosurid had undergone synchrotron CT scanning. So, the four specimens scanned by the authors definitely represent a welcome addition. The method and its research potential are sufficiently explained, and this part definitely meets the requirements set by PeerJ.

Validity of the findings

Some of the discussion and conclusions are a bit muddled, in part due to rather imprecise phrasing, which makes it unclear what exactly the authors are trying to say (e.g. "cuticularised", "ventral preservation"). Also, while the text states that the CT imaging revealed moveable spines in Victalimulus mcqueeni, only the fixed spines, and intervening notches are visible in the images provided. For a detailed discussion of these issues, please see the "Additional comments", and the annotated manuscript.

Additional comments

Review of “An earliest Triassic age for Tasmaniolimulus and comments on synchrotron tomography of Gondwanan horseshoe crabs” (#67956)

Horseshoe crabs are an iconic clade of euarthropods with a long evolutionary history, and the only extant fully aquatic chelicerates. While currently only four species are in existence, their taxonomic diversity and morphological disparity used to be much greater in the Paleozoic and Mesozoic. Therefore, any paper dealing with fossil xiphosurids is of considerable interest to palaeobiologists and invertebrate and evolutionary biologists.

The current manuscript revises the age of Tasmaniolimulus, previously believed to be Permian, to the Triassic, which has implications for our understanding of the timing of the radiation of the family Austrolimulidae. In addition, it discusses the results of synchrotron CT scanning of four specimens of fossil Australian horseshoe crabs. The extremely high energies, brilliance and almost monochromatic character of synchrotron light allow a penetration and resolution unequalled by classical X-ray tomography. Therefore, synchrotron tomography is quickly becoming established as an extremely valuable technique in the study of fossil specimens. However, so far, synchrotron tomography had only been applied to one fossil horseshoe crab specimen. The four specimens scanned for this paper therefore represent a welcome addition, and provide some additional morphological data over what was known previously from more classical studies of the material using imaging in visible wavelengths. Consequently, this paper is of interest, and deserves to be published. There are, however, a number of issues that need to be addressed first.

On p. 14, line 221, the authors state that “... a limited portion of the spine was cuticularised.” What is this supposed to mean? What is “cuticularised”? Xiphosurids are euarthropods. Euarthropods are Ecdysozoa. Ecdysozoans, as one of their primary defining characters, have an external cuticle. In euarthropods, most of this cuticle (except for arthrodial membrane) is at least to some degree sclerotised to form an exoskeleton. So, a euarthropod, by definition, is externally encased in a sclerotised cuticle. I suppose the authors may be trying to say to say that the degree of sclerotisation varied along the spine, or that the whole spine was less sclerotised compared to the rest of the prosomal shield? This sentence does need rephrasing to make sense – talking about a portion of the external surface of an ecdysozoan being “cuticularised” is akin to talking about a portion of a mammalian spine being “notochordised”.

On p. 15, lines 227-228, it is stated that “The reconstruction of Victalimulus mcqueeni reveals evidence for the thoracetronic doublure, moveable spines and notches, and appendage impressions.”
- First, after reviewing both Fig. 4 and the supplemental image, I fail to see the moveable spines – I can only make out the fixed spines of the thoracetron, and the intervening notches that accommodated the moveable spines, not the moveable spines themselves. Indeed, the caption to Fig. 4 itself states: “... X-ray tomographic slice showing fixed spines and moveable spine notches (white arrows) and thoracetronic doublure”; so the caption itself does not mention the presence of the actual moveable spines either. If the moveable spines are indeed visible anywhere, they should be indicated.
- Second, the statement is somewhat misleading, considering that the original description by Riek & Gill (1971) already commented on the thoracetronic doublure, the presence of fixed spines and notches, and the impressions of the prosomal walking limbs. Therefore, these do not exactly represent new findings.

On p. 16, the authors talk about “dorsal preservation”, and “ventral preservation”. It is not entirely clear what they mean by this:
- Does it mean that only the dorsal (or ventral) exoskeleton is preserved?
- Or, do the authors intend to say that the fossils are exposed from the dorsal (or ventral) side?
- Or, do they intend to convey that the original orientation in which the specimens were preserved was with the dorsal (or ventral) side up?
These are not trivial differences in meaning. The authors comment that “dorsal preservation” is more common than “ventral preservation” in horseshoe crabs. If they are referring to the surface of the fossil that is exposed, this is not surprising: the dorsal surface represents a relatively smooth, convex surface, whereas the concave ventral surface is much less regular. As such, the dorsal surface will represent a plane of weakness along which the matrix containing the fossil will preferentially fracture. It therefore makes sense for the majority of xiphosurid specimens to be exposed from the dorsal side – as is observed for most other dorso-ventrally flattened euarthropods. However, the fact that these specimens are exposed on the dorsal side does not necessarily mean that they have been preserved dorsal side up: it is common for specimens which have been embedded dorsal side down to still crack out along the dorsal surface, because of the aforementioned reasons. Hence, if it is the intention of the authors to make the case that most horseshoe crab fossils were preserved with the dorsal side up, they need to provide additional arguments for the original orientation of the specimen – either from sedimentological characteristics of the surrounding matrix that allow polarity to be deduced, or from data documenting the original orientation of the fossil when collected.

On the same page, the authors also comment that “... a ventral orientation has a lower preservational protentional”. Again, if they are just referring to the side of the fossil that is exposed, it makes sense, as explained above, that the majority of fossils exposes the dorsal side. However, if the argument is that specimens being preserved with their ventral side up stand a lesser chance of preservation, this also makes sense: the ventral exoskeleton is considerably less sclerotised than the dorsal side. So, if a carcass is lying on the sea floor upside down, with its ventral side exposed, scavengers and currents will have free reign to tear it up. If, on the other hand, the carcass is not overturned, the heavy dorsal exoskeleton will to some extent help protect the more fragile ventral anatomy from being destroyed, allowing time for the carcass to be buried, potentially helping it to enter the fossil record. In this respect, a dorsal-side-up orientation may actually aid preservation of limbs and soft anatomy, rather than “damage” it, as the authors suggest. In that case, the overlying dorsal exoskeleton would obscure any underlying preserved parts, making it harder for them to be studied, but in itself would not “damage” them; in this respect, it is also important to note that at least in the case of non-biomineralising arthropods, flattening of carcasses mostly results from decay-induced collapse of the carcass onto itself, rather than from compaction by the surrounding sediment.

On p. 19, lines 317-319, it is claimed that the Winterswijk Limulitella shows “extensive soft tissue traces”. At least based on the information provided in the paper by Zuber et al. (2017), this claim is entirely incorrect: the authors of that study only identified exoskeletal morphological features of this fossil. They do refer to the presence of “muscular markings”, but as is clear from their figures, by this they do not mean actual muscle tissue, but rather the depressions on the prosomal shield associated with internal apodemes to which muscles would have attached. Hence, the only thing visible in their scans was the sclerotised, exoskeletal cuticle, which is neither soft, nor a tissue!

Apart from these issues, I have added some further small remarks and (mainly typographical) corrections as sticky notes to the attached annotated manuscript.

In conclusion, this manuscript does present some interesting data, and does merit to be published in PeerJ. However, before it can be published, the authors do need to rectify the several issues raised here.

Peter Van Roy
23 December 2021

---

## Round 0.2 · Minor Revisions

Dear authors,

Thank you for your careful changes based on the previous round of reviews. Reviewer 2 has noted a few minor changes that should be incorporated into the manuscript and then I believe that it will be publishable in PeerJ. Particularly, noting the utility of non-synchrotron based surface methods for generating subtle surface features, such as surface scanning (which would likely not have the same resolution) should be discussed.

I caught a few minor typographical errors too upon my last reading. When you resubmit your paper, please include a tracked changes version, clean version, and response to reviewers document. Thank you for your submission to PeerJ. Please let me know if you have any questions.

Best,

Brandon P. Hedrick, Ph.D.


Line 301: I think it should be a comma before ‘CFD studies’ rather than a period

Line 302: There’s a shift from ‘CFD studies’ to ‘CFDA studies’. Either way works, but probably change it on line 302 for consistency with the rest of the paper.

Line 314: CDF should be CFD here

·

Basic reporting

I am fine with the changes that were made.
I only found one issue that needs to be checked: The section "Comments on application of synchrotron tomography to the study of fossil xiphosurids" in the discussion is not entirely the same in the PDF and in the DOCX. In the DOCX, this header appears two times, and then parts of the text are different in the two files. The authors need to have a close look here again to make sure that they submit the correct final version.

Experimental design

none

Validity of the findings

none

Additional comments

none

·

Basic reporting

No comment.

Experimental design

No comment.

Validity of the findings

No comment.

Additional comments

The authors have made a genuine effort to accommodate the queries, suggestions and criticisms of both reviewers, which is appreciated. As a result, this revised manuscript is considerably better than the original version. Nevertheless, there are still some remaining issues the authors need to address before the paper can be accepted for publication:

- Both the abstract and Results section claim that the SRXT images revealed novel anatomical details beyond what was already known. In truth, this is an exaggeration: no genuinely new morphological/anatomical information was revealed; the SRXT scans only served to further emphasize and document some morphological details that were already known previously. I therefore strongly suggest the authors tone down their statements about the novelty of the findings to reflect this - indeed, in their concluding remarks, the authors themselves admit that the current study did not reveal 'much novel anatomy, nor traces of soft tissues'!

- The introduction still refers to 'ventral preservation' of specimens. As I discussed at length in my earlier review of the original manuscript, this term is ambiguous at best, and the authors should rephrase this to make clear what they mean exactly.

- In several places, the name of the Anisian stage of the Triassic is rendered as 'Ansian', dropping the first 'i'. This should be checked and corrected throughout the manuscript.

- The main purpose of CT and SRXT scanning is to reveal internal features of a specimen, or show features hidden in the matrix. This study did not really accomplish either of those goals, but did result in high-resolution 3D surface models of the specimens, which is worthwhile in and by itself. However, 3D digitalization of specimens in this way can be accomplished more easily with e.g. laser scanning, or to some extent even with photogrammetry and RTI imaging (although the latter two techniques have considerably lower resolution, and tend to suffer from artefacts, in particular in highly reflective areas). Therefore, the authors may consider adding a couple of words about these techniques to their discussion of the 3D digitalization of specimens.

- In a couple of places, 'parvinospinoslis' should read 'parvinospinosus'. Check!

- Moveable notches: sloppy terminology! The notches are not moveable - they accommodate moveable spines (which are not preserved in the specimen).

- Use Konservat-Lagerstätte instead of just 'Lagerstätte'. Seilacher distinguished two main types of Lagerstätten: Konservat-Lagerstätten, which are sites with exceptional preservation, and Konzentrat-Lagerstätten, which are sites characterized by an abundance of fossil material. Therefore, it is important to distinguish between the two - using just Lagerstätte makes it unclear what type of deposit you are referring to.

In addition to these remarks, I have made some smaller comments and (mostly typographical) corrections on the attached manuscript.

In summary, this revised manuscript is a marked improvement of the original, and if the authors address the remaining issues flagged here, the paper should be acceptable for publication.

Peter Van Roy
02 March 2022

---

## Round 0.3 · accepted · Accept

Dear authors,

Thank you for your careful changes based on the previous round of reviews. I now find this manuscript to be publishable in PeerJ and want to move it to the next stage. Congratulations!

There were just a few small grammatical points that should be fixed prior to publication:

Line 130: space before final period of sentence

Line 142: Stereo-photographs don’t get internal anatomy either. Perhaps “However, neither photogrammetry (nor stereo-photographs) can be used to…”

Line 188-189: ‘…Basin, which suggest…’


Please contact me if you have any questions. Thank you for your submission to PeerJ.

Brandon P. Hedrick, Ph.D.